# Effects of Combined Live *Bifidobacterium*, *Lactobacillus*, *Enterococcus*, and *Bacillus* Cereus Tablets on the Structure and Function of the Intestinal Flora in Rabbits Undergoing Hepatic Artery Infusion Chemotherapy

**DOI:** 10.3390/biology13050327

**Published:** 2024-05-08

**Authors:** Xiangdong Yan, Liuhui Bai, Jin Lv, Ping Qi, Xiaojing Song, Lei Zhang

**Affiliations:** 1The First Clinical Medical College, Lanzhou University, Lanzhou 730000, China; yanxd21@lzu.edu.cn (X.Y.); bailh21@lzu.edu.cn (L.B.); lvj21@lzu.edu.cn (J.L.); qip21@lzu.edu.cn (P.Q.); songxiaojing4227@126.com (X.S.); 2Department of General Surgery, The First Hospital of Lanzhou University, Lanzhou 730000, China; 3Key Laboratory of Biotherapy and Regenerative Medicine of Gansu Province, The First Hospital of Lanzhou University, Lanzhou 730000, China

**Keywords:** adverse reactions to chemotherapy drugs, intestinal microorganisms, intestinal barrier, probiotics, Combined Live *Bifidobacterium*, *Lactobacillus*, *Enterococcus*, and *Bacillus* Cereus Tablets

## Abstract

**Simple Summary:**

Patients with hepatocellular carcinoma (HCC) often have a variety of adverse reactions after chemotherapy, which may be related to the changes in intestinal flora caused by chemotherapy. In this study, a rabbit model of liver local arterial perfusion chemotherapy was established. The effects of Combined Live *Bifidobacterium*, *Lactobacillus*, *Enterococcus*, and *Bacillus* Cereus Tablets on the structure of intestinal microbiota and intestinal barrier function in rabbits treated with local arterial perfusion chemotherapy were studied, and the possible potential mechanisms were discussed. This study provides a theoretical basis for probiotics to regulate intestinal microbiota and improve the side effects of local arterial perfusion chemotherapy drugs in the liver. After arterial chemotherapy with epirubicin, the relative abundance of beneficial bacteria, such as *Cladosporium and Bacteroides*, decreased, while the relative abundance of harmful bacteria, such as *Proteus*, increased. In rabbit serum and ileum tissue, the protein expression levels of the intestinal barrier tight-junction proteins ZO-1, occludin, and claudin-1 decreased, while the levels of inflammatory factors and liver injury factors increased. Combined Live *Bifidobacterium*, *Lactobacillus*, *Enterococcus*, and *Bacillus* Cereus Tablets can reverse the composition of intestinal flora. The relative abundance of beneficial bacteria, such as rabbit *Bifidobacterium*, rumen coccus, Prevost’s bacteria, and *Blautia*, increased, while the relative abundance of *Proteus*, *verrucous microorganisms*, *Escherichia/Shigella*, and *Sporobacter* decreased. It also significantly increased the expression of the rabbit tight-junction proteins ZO-1, occludin, and claudin-1, increased the expression of mucin gene, reduced intestinal permeability, and repaired intestinal barrier injury. Probiotic regulation reduces the expression levels of inflammatory factors and liver injury factors in rabbit serum. The correlation analysis between intestinal flora and clinical environmental factors showed that beneficial bacteria were positively correlated with the protein expression levels of the intestinal tight-junction proteins claudin-1, ZO-1, and occludin, while beneficial bacteria were negatively correlated with harmful bacteria. The expression level of beneficial bacteria was negatively correlated with inflammatory factors and liver injury factors, and positively correlated with harmful bacteria.

**Abstract:**

Few studies have explored the biological mechanism by which probiotics alleviate adverse reactions to chemotherapy drugs after local hepatic chemotherapy perfusion by regulating the intestinal flora. This study investigates the effects of Combined Live *Bifidobacterium*, *Lactobacillus*, *Enterococcus*, and *Bacillus Cereus* Tablets on the intestinal microbial structure and intestinal barrier function, as well as the potential mechanism in rabbits after local hepatic chemotherapy infusion. Eighteen New Zealand White rabbits were randomly divided into a control group, a hepatic local chemotherapy perfusion group, and a hepatic local chemotherapy perfusion + Combined Live *Bifidobacterium*, *Lactobacillus*, *Enterococcus*, and *Bacillus* Cereus Tablets group to assess the effects of Combined Live *Bifidobacterium*, *Lactobacillus*, *Enterococcus*, and *Bacillus* Cereus Tablets on the adverse reactions. The administration of Combined Live *Bifidobacterium*, *Lactobacillus*, *Enterococcus*, and *Bacillus* Cereus Tablets alleviated the intestinal flora disorder caused by local hepatic perfusion chemotherapy, promoted the growth of beneficial bacteria, and inhibited the growth of harmful bacteria. The Combined Live *Bifidobacterium*, *Lactobacillus*, *Enterococcus*, and *Bacillus* Cereus Tablets also reduced the levels of serum pro-inflammatory cytokines and liver injury factors induced by local hepatic perfusion chemotherapy. Our findings indicate that Combined Live *Bifidobacterium*, *Lactobacillus*, *Enterococcus*, and *Bacillus* Cereus Tablets can ameliorate the toxicity and side effects of chemotherapy by regulating intestinal flora, blocking pro-inflammatory cytokines, reducing liver injury factors, and repairing the intestinal barrier. Probiotics may be used as a potential alternative therapeutic strategy to prevent the adverse reactions caused by chemotherapy with local hepatic perfusion.

## 1. Introduction

Liver cancer is one of the most common malignant tumors worldwide, with the fourth-highest incidence rate and third-highest mortality rate [1]. The early symptoms of patients with primary liver cancer are not obvious, and many patients are in advanced stages of liver cancer at the time of visiting a doctor, thus losing the opportunity for surgical treatment. Transarterial chemoembolization (TACE) is the preferred non-surgical treatment for patients with advanced hepatocellular carcinoma [2]. Chemotherapy remains one of the best treatments for patients with advanced liver cancer at present; however, adverse reactions during treatment seriously impact the therapeutic effect and prognosis of these patients and negatively affect their quality of life.

Gastrointestinal toxicity is one of the most common and severe adverse reactions of chemotherapy, with the main manifestations being intestinal mucositis, diarrhea, and constipation [3,4,5]. Many studies have shown that the intestinal flora may affect the efficacy and toxicity of chemotherapeutic drugs, and that the imbalance of the intestinal flora is related to changes in the intestinal mucosa and mucosal inflammation. The intestinal flora plays a vital role in chemotherapy and immunotherapy, as well as in gastrointestinal toxicity and side effects [6]. The regulation of the intestinal microbiota may increase the effectiveness of many cancer treatments while reducing cytotoxic activity [7]. Although different chemotherapeutic drugs may have different effects [8,9], overall, they decrease the abundance of lactic acid bacteria and *Bifidobacteria* and increase the abundance of *E. coli* and *Staphylococcus* [10]. Therefore, intestinal microbiota can be manipulated using antibiotics, probiotics, prebiotics, or fecal transplantation to improve the efficacy and reduce the toxicity of chemotherapeutic drugs [11,12].

Probiotics can reduce the micro-inflammatory reaction of the intestinal mucosa during chemotherapy and maintain the stability of the intestinal barrier function, which can reduce the incidence of adverse reactions caused by chemotherapy drugs [13,14,15]. Few studies have systematically analyzed the effect of oral probiotics on chemotherapy-induced mucositis in patients with liver cancer or evaluated the changes in intestinal flora of patients with liver cancer after probiotic treatment. Therefore, this study established a rabbit model of liver local chemotherapy perfusion. We studied the influence of Combined Live *Bifidobacterium*, *Lactobacillus*, *Enterococcus*, and *Bacillus* Cereus Tablets on the intestinal microbial structure and intestinal barrier function of model rabbits, and provided a theoretical basis for probiotics to reduce the toxicity and side effects of local chemotherapy drugs in the liver.

## 2. Materials and Methods

### 2.1. Experimental Materials

Eighteen normal, healthy, male New Zealand white rabbits, weighing 2.5–3.5 kg, were selected. In accordance with the National Animal Care and Use Committee, they were raised at the Lanzhou University School of Medicine Experimental Animal Center under standard conditions. Rabbits were anesthetized by intravenous injection of 30% pentobarbital sodium (20,220,125 mg/kg) through the ear margin. The animal program was approved by the Ethics Committee of the First Hospital of Lanzhou University (approval number: LDYYLL2024-12).

### 2.2. Experimental Operation

Eighteen New Zealand white rabbits were randomly divided into two groups: the normal control group (*n* = 6) and the experimental group (*n* = 12). In the experimental group, 1% pentobarbital sodium (30 mg/kg body weight) was administered by a slow static push through the auricular vein. After the disappearance of the corneal reflex, an incision was made in the middle of the abdomen for laparotomy to identify the inherent hepatic artery. Under direct vision, 10 mg of epirubicin (Shenzhen Meile Pharmaceutical Co., Ltd., Shenzhen, China) was mixed with 1 mL of normal saline using a syringe. The mixture was injected into the hepatic artery of the experimental rabbits at a dose of 0.15 mL/kg and sutured layer-by-layer. After the successful establishment of the model, the experimental group was randomly divided into two groups. The first was the model group, or the liver local chemotherapy perfusion group (*n* = 6). In this paper, it is collectively called the model group, and in the figures, “model group” is uniformly used for labeling. The second was the probiotic intervention group: liver local chemotherapy and perfusion combined with Combined Live *Bifidobacterium*, *Lactobacillus*, *Enterococcus*, and *Bacillus* Cereus Tablets (*n* = 6). In this paper, this group is collectively referred to as the probiotic intervention group, and in the figures, the “probiotic group” is uniformly used for labeling. The rabbits in the normal control and liver local chemotherapy perfusion groups (*n* = 6) were intragastrically administered the same dose of sterile normal saline. After the rabbit model was successfully established by local liver chemotherapy perfusion, the probiotic intervention group consumed 0.75 mg/bid of Combined Live *Bifidobacterium*, *Lactobacillus*, *Enterococcus*, and *Bacillus* Cereus Tablets (2022070565, Hangzhou Yuanda Biology Pharmacy Co., Ltd., Hangzhou, China) for 7 days. After 7 days of intervention, rabbits were anesthetized with 30% pentobarbital sodium. After the abdominal cavity was opened, the rabbit abdominal aorta was searched for blood collection under direct vision. Then, the ileal tissues and ileal feces were collected into cryotubes and stored in a −80 °C refrigerator. The changes in intestinal flora were analyzed by 16S rRNA sequencing, and the expression levels of serum AST, ALT, DAO, pNF-κB, p-STAT3, IL-6, LPS, TNF-α, COX-2, GM-CSF, and NLRP3 were detected using the enzyme-linked immunosorbent assay (ELISA). The serum D-lactic acid levels were detected using the D-lactic acid colorimetric method. Moreover, the gene expression of claudin-1, ZO-1, and occludin in the ileum and of claudin-1, ZO-1, occludin, and MUC2 in the serum were detected using ELISA.

### 2.3. Biological Analysis of Intestinal Flora in Rabbits

#### 2.3.1. Sample Processing and DNA Extraction of Rabbit Feces

Firstly, the collected rabbit feces samples were frozen in a liquid nitrogen tank and immediately transferred to the refrigerator at −80 °C for storage until analysis. According to the instructions of the kit manufacturer, we extracted genomic DNA from rabbit feces by using the QIAam P®DNA Fecal Mini Kit (Qiagen, Hilden, Germany). We detected the concentration and purity of rabbit feces samples using nano-drop technology. The integrity of rabbit feces samples was detected using routine 0.8% agarose gel electrophoresis.

#### 2.3.2. Rabbit Intestinal Flora High-Throughput Sequencing

Using rabbit bacterial genomic DNA as template, we amplified the V3-V4 hypervariable region of the 16S rRNA gene by using forward and reverse primers. Among them, the RNA sequence of the forward primer was 5′-CCTACGGGGCWGCAG-3′, and the RNA sequence of the reverse primer was 5′-GACTACHVGGGTATCTAATCC-3′. Each rabbit stool sample was amplified independently 3 times. In the end, the products of polymerase chain reaction (PCR) were detected by agarose gel electrophoresis and collected with the PCR products of the rabbit feces samples. We used the polymerized PCR products as templates, used index primers for exponential PCR, and added the Illumina index to the library. Next, we verified the sequence of PCR amplification products by gel electrophoresis. Then, the PCR amplification sequence was purified via the AgcurtAMPureXP Kit (Beckman Coulter, Brea, CA, USA). In the 16SV3-V4 library, the purified PCR amplification sequence products were detected. We used a Qubit@2.0 Fluorometer (Thermo Scientific, Waltham, MA, USA) and the Agilent Bioanalyzer 2100 system to evaluate the library quality. Finally, the assembled library was sequenced on the Illumina MiSeq 250 Sequencer, and 2 × 250 bp opposite reads were obtained.

#### 2.3.3. Biological Analysis of Rabbits

Standards for filtering and combining the reading quality of raw data of rabbit biological information were as follows: (1) Truncate the original data reading with an average quality score of less than 20 at any position, delete the reading polluted by the adapter, and further delete the reading by less than 100 basis points. (2) Adjust the length of the pairing reading by using short reading (FLASH, v1.2.11). (3) Then, remove vague bases (N bases) and homopolymers reading more than 6 bp and, finally, (4) remove the reading with low complexity to obtain a clean reading. By comparing with the gold.fa database (http://drive5.com/uchime/gold.fa accessed on 2 March 2024), the other unique reads were checked for mosaic, and UPARSE clustering with a similarity cut-off of 97% was used as the operation classification unit. According to Mothur’s ribosome database project (RDP), release 9 201203, all OTUs were classified. The sparsity and alpha diversity (including the Shannon index, Simpson index, and Jing–Simpson index) were analyzed using Mothur. Using R project (Vegan Software Package, V3.3.1), the sample tree was clustered using the Bray–Curtis distance matrix. The original data were analyzed using the unweighted average (UPGMAO), weighted, Curtis, and Yaka principal coordinate analysis (PCoA) based on OTUs. Redundancy analysis (RDA) was conducted for Windows 4.5 (Microcomputer Power Supply, New York, NY, USA) using Canoco, and 499 random permutations were evaluated using MCPP.

### 2.4. ELISA

The following kits were used: the rabbit tight-junction protein (occludin) ELISA assay kit (ml005521, Shanghai Enzyme-linked Biotechnology, Shanghai, China); rabbit tight-junction protein 1 (ZO-1) ELISA kit (ml027943, Shanghai Enzyme-linked Biotechnology, Shanghai, China); rabbit occludin 1 (claudin-1) ELISA detection kit (ml666985, Shanghai Enzyme-linked Biotechnology, Shanghai, China); rabbit lipopolysaccharide (LPS) ELISA Kit (ml44523, Shanghai Enzyme-linked Biotechnology, Shanghai, China); rabbit diamine oxidase (DAO) ELISA kit (ml027888, Shanghai Enzyme-linked Biotechnology, Shanghai, China); rabbit secretory immunoglobulin A (sIgA) kit (ml036798, Shanghai Enzyme-linked Biotechnology, Shanghai, China); rabbit interleukin-10 (IL-10) kit (ml332541, Shanghai Enzyme-linked Biotechnology, Shanghai, China); rabbit interleukin-6 (IL-6) ELISA detection kit (ml125512, Shanghai Enzyme-linked Biotechnology, Shanghai, China); rabbit tumor necrosis factor α (TNFα) ELISA kit (ml027766, Shanghai Yuanju Biotechnology, Shanghai, China); rabbit aspartate aminotransferase (AST) kit (ml036743, Shanghai Yuanju Biotechnology, Shanghai, China); rabbit alanine aminotransferase (ALT) ELISA reagent (ml027206, Shanghai Yuanju Biotechnology, Shanghai, China); rabbit granulocyte macrophage colony-stimulating factor (GM-CSF) ELISA detection kit (ml421125, Shanghai Yuanju Biotechnology, Shanghai, China); rabbit NLR family Pyrin domain protein 3 (NLRP3) ELISA detection kit (ml665211, Shanghai Enzyme-linked Biotechnology, Shanghai, China); rabbit mucin 2 (MUC2) ELISA detection kit (ml555632, Shanghai Enzyme-linked Biotechnology, Shanghai, China); rabbit phosphorylated nuclear transcription factor (pNF-κB) ELISA detection kit (ml459985, Shanghai Enzyme-linked Biotechnology, Shanghai, China); rabbit phosphorylation signaling and activator of transcription 3 (p-STAT3) ELISA detection kit (ml1144586, Shanghai Enzyme-linked Biotechnology, Shanghai, China); rabbit cyclooxygenase 2 (COX-2) ELISA detection kit (ml3332512, Shanghai Enzyme-linked Biotechnology, Shanghai, China); rabbit Toll-like receptor 4 (TLR4) ELISA detection kit (ml2254156, Shanghai Enzyme-linked Biotechnology, Shanghai, China).

### 2.5. D-Lactic Acid Colorimetric Detection

A rabbit D-lactic acid colorimetric assay kit (ml622365; Shanghai Yuanju Biotechnology, Shanghai, China) was used.

### 2.6. Statistical Analysis

All statistical analyses were performed using SPSS 22.0 for Windows. To determine the statistical differences between two groups, we used an independent samples *t*-test and the Mann–Whitney U test. Correlations were assessed using Spearman’s correlation coefficients. The resulting *p*-values were adjusted using the Benjamini–Hochberg false discovery rate (FDR) correction. Only FDR-corrected *p*-values < 0.05 were considered significant.

## 3. Results

### 3.1. Effects of Combined Live Bifidobacterium, Lactobacillus, Enterococcus, and Bacillus Cereus Tablets on the Overall Structure of the Intestinal Flora

#### 3.1.1. Alpha Diversity Analysis of Rabbit Intestinal Flora

Alpha diversity refers to the community diversity mainly used to assess the intestinal microorganisms in each group. The flora composition in the sample group was assessed by calculating a series of alpha diversity indices. In addition, a statistical *t*-test was applied to the α diversity index among different groups to test whether the index values had significant differences. In this study, we used different diversity indicators to evaluate the diversity of intestinal microorganisms in rabbits (Observed, Chao, ACE, Shannon, Simpson, and Coverage indices). As shown in Figure 1, compared with the control group, there was no significant change in the intestinal microbial diversity and α diversity in the experimental and probiotic intervention groups. Compared with the control group, the microbial diversity of the intestinal flora in the model group decreased, and the diversity of the intestinal flora changed a little after acute intervention with probiotics. The Observed, Chao, ACE, and Shannon indices were higher in the model group, whereas opposite changes were observed regarding the Simpson diversity index. We found that there was no significant difference in the α diversity among the different groups when the diversity of microbial communities was analyzed.

#### 3.1.2. Beta Diversity Analysis of Rabbit Intestinal Flora

The β diversity refers to the comparative analysis of the composition of intestinal flora in different groups of samples. As shown in Figure 2, there were significant differences in the composition of intestinal flora in each group. Figure 2A presents the PCoA diagram, which shows that there were obvious differences in microbial communities among groups. There was a significant difference between the control group and the model group. There was also a significant difference between the model group and the probiotic intervention group. Figure 2B presents the PLS-DA diagram, which shows significant differences in bacterial community structure among groups.

#### 3.1.3. Combined Live *Bifidobacterium, Lactobacillus, Enterococcus*, and *Bacillus* Cereus Tablets Improved the Intestinal Flora Composition of Rabbits Undergoing Local Hepatic Perfusion Chemotherapy

As shown in Figure 3 and Figure 4, we found that there were significant differences in the relative abundances at the phylum, genus, family, and species levels in the intestinal flora of rabbits in the control, experimental, and probiotic intervention groups. The viable probiotic bacteria, *Bifidobacterium* bifidum, restored the overall structure of the intestinal flora by restoring the imbalance in the intestinal flora caused by chemotherapy drugs. Figure 3A shows that at the phylum level, the relative abundance of *Proteobacteria and Verrucomicrobia* in the model group increased, while the relative abundance of *Firmicutes and Bacteroides* decreased. Compared with the model group, after the probiotic intervention of Combined Live *Bifidobacterium*, *Lactobacillus*, *Enterococcus*, and *Bacillus* Cereus Tablets, the relative abundances of rabbit *Firmicutes*, *Bacteroides*, and *Actinobacteria* increased, while the relative abundances of *Proteobacteria* and *Verrucomicrobia* decreased.

Figure 3B shows that at that the family level, in the model group, the relative abundance of *Enterobacteriaceae*, *Verrucomicrobiacea,* and *Desulfovibrionaceae* increased, while the relative abundance of *Bacteroideae* and *Porphyromonadaceae* decreased. Compared with the model group, after the probiotic intervention, the relative abundance of *Bacteroidaceae*, *Actinobacteria*, *Christensenellaceae*, *Bifidobacteriae*, *Erysipelotrichaceae,* and *Prevotellaceae* increased, whereas that of *Enterobacteriae*, *Verrucomicrobiaceae*, and *Desulfovibrionaceae* decreased.

Figure 3C shows that at the genus level, the relative abundance of *Escherichia/Shigella*, *Akkermansia*, and *Sporobacter* in the model group increased, while the relative abundance of *Bacteroides* and *Blautia* decreased. Compared with the model group, after the probiotic intervention, the relative abundance of *Bacteroides*, *Blautia,* and *Bifidobacterium* increased, while the relative abundance of *Escherichia/Shigella* and *Sporobacter* decreased.

Figure 3D shows that at the species level, in the model group, the relative abundance of *Clostridiales*, *Proteobacteria*, *Desulforivibrios*, and *Verrucomicrobiales* increased, while the relative abundance of *Bacteroides* and *Lactobacillales* decreased. Compared with the model group, after the probiotic intervention, the relative abundance of *Bacteroides*, *Lactobacillus*, *Bifidobacterium,* and *Lactobacillus* in rabbits increased, while the relative abundance of *Proteobacteria*, *Verrucomicrobiales*, and *Desulforivibrios* decreased.

In Figure 4, the linear discriminant analysis (LDA) effect quantity method (LEfSe) was used as the difference screening threshold to select the species that were most likely to explain the differences among groups. These species are considered as potential biomarkers of various populations. Because of the obvious changes in intestinal flora in each group, we analyzed the species composition in each group. Through LEfSe, we observed the significant differences of species enriched in the model group and the probiotic intervention group. By analyzing the LDA score, the significantly enriched flora in the model group were *p_Proteobacteria*, *c__Gammaproteobacteria*, *g_Erysipelotrichaceae_incertae_sedis*, *g_Clostridium_XlVa*, *g__Bilophila*, *g__Aestuariispira*, *f__Rhodospirillaceae*, and *o__Rhodospirillales.* In the probiotic intervention group, the enriched flora were *g__Bacteroides*, *c__Erysipelotrichia*, *g__Oscillibacter*, *s__Christensenella_minuta*, *g__Christensenella*, *s__uncultured_organism*, *f__Christensenellaceae*, *g__Hungatella*, *f__Erysipelotrichaceae*, and *o__Erysipelotrichales*.

### 3.2. Effects of Combined Live Bifidobacterium, Lactobacillus, Enterococcus, and Bacillus Cereus Tablets on the Intestinal Barrier Function of Rabbits

#### 3.2.1. The Probiotics Significantly Enhanced the Expression of Tight Junction Proteins in the Intestinal Barrier of Rabbits

We detected the protein expression of the intestinal barrier tight-junction proteins (claudin-1, ZO-1, and occludin) in the ileum and the protein expression of claudin-1, ZO-1, occludin, and MUC2 in the serum using ELISA. As shown in Figure 5, in the serum, compared with the control group, the expression levels of claudin-1, ZO-1, and occludin in the model group were significantly lower (*p* < 0.01). The protein expressions of the intestinal barrier proteins (claudin-1, ZO-1, and occludin) were restored after regulation by probiotics (*p* < 0.01). Regarding MUC2, the expression was reduced in the model group compared with the control group, and the serum expression of MUC2 was increased after the probiotic intervention. In the intestinal tissues, compared with the control group, the protein expression levels of claudin-1 and ZO-1 were significantly decreased in the model group (*p* < 0.01), and that of occludin was also decreased (*p* < 0.05). After the probiotic intervention, the expression levels of claudin-1, ZO-1, occludin, and MUC2 were significantly increased (*p* < 0.05), which were not significantly different from those in the control group but were significantly different from those in the model group.

#### 3.2.2. The Probiotics Reduced the DAO and D-Lactic Acid Levels (Indicators of Intestinal Barrier Permeability of Rabbits)

Increased intestinal permeability resulted in increased serum levels of DAO and D-lactic acid; therefore, the serum levels of DAO and D-lactic acid were responsive to changes in intestinal permeability. The effects of the probiotics on rabbit serum DAO and D-lactic acid levels are shown in Figure 6. Compared with the control group, the DAO and D-lactic acid levels in the experimental group were significantly higher (*p* < 0.01). After probiotic regulation, the DAO and D-lactic acid levels decreased, which could have reduced the intestinal permeability and repaired the intestinal barrier, to a certain extent.

### 3.3. Effects of the Probiotics on Serum Clinical Inflammatory Factors in the Rabbits

#### 3.3.1. The Probiotics Reduced Rabbit Serum Inflammatory Factors

To determine the effects of probiotics on the inflammatory factors, such as LPS, TNF-α, NF-κB, and IL-6, in rabbits undergoing local hepatic perfusion chemotherapy, ELISA was used to detect the content of serum inflammatory factors. The test results are presented in Figure 7. Compared with the control group, the serum contents of LPS, TNF-α, and IL-6 in rabbits in the model group were significantly increased (*p* < 0.01). Compared with the model group, the regulated serum expression levels of TNF-α and IL-6 in the probiotic group were significantly decreased (*p* < 0.05). After regulation, the serum expression levels of LPS and pNF-κB in the probiotic group decreased, all of which were higher than those in the control group.

#### 3.3.2. Regulation of the Intestinal Flora by Probiotics to Reduce Inflammatory Factors of Liver Injury

To explore the effects of probiotics on liver injury inflammatory factors, such as ALT, AST, NLRP3, and STAT3, in rabbits following local hepatic perfusion chemotherapy, ELISA was used to assess the content of liver injury inflammatory factors in rabbit serum. The detection results are presented in Figure 8. Compared with the control group, the levels of ALT, AST, NLRP3, and STAT3 in the rabbit serum of the model group were significantly higher (*p* < 0.01). After regulation, the expression levels of ALT and STAT3 in the serum of rabbits in the probiotic group significantly decreased (*p* < 0.05). Moreover, after regulation, the expression levels of ALT and NLRP3 in the serum of rabbits in the probiotic group decreased and were all higher than those in the control group.

### 3.4. Correlation between the Structure and Function of the Intestinal Flora and Clinical Inflammatory Factors in the Rabbits

To explore the relationship between the structure and function of rabbit intestinal flora and clinical inflammatory factors, a correlation coefficient was used to analyze the relationship between probiotics and clinical environmental factors. As shown in Figure 9A, at the phylum level, *Firmicutes* exhibited a significant positive correlation with ZO-1 and occludin expression (*p* < 0.01). There was a significant positive correlation with the expression of the intestinal barrier proteins ZO-1 and occludin (*p* < 0.01) and a positive correlation with the expression of claudin-1 (*p* < 0.05). *Proteobacteria* had a significant positive correlation with LSP, AST, ALT, TNF-α, and p-STAT3 (*p* < 0.01). Moreover, it had a positive correlation with DAO and IL-6 levels (*p* < 0.05).

Figure 9B shows that at the genus level, the harmful bacterium *Escherichia/Shigella* had a significant positive correlation with the LSP and TNF-α levels (*p* < 0.01) and a positive correlation with the AST, ALT, and COX-2 levels (*p* < 0.05). However, there was a significantly negative correlation with the intestinal barrier proteins claudin-1, ZO-1, and occludin (*p* < 0.01). *Lachnospiraceae_incertae_sedis* was significantly positively correlated with the intestinal barrier proteins ZO-1 and occludin (*p* < 0.01). Moreover, it was positively correlated with the expression of claudin-1 (*p* < 0.05). There was a strong negative correlation between *Anaerobacteria* and the expression levels of claudin-1, ZO-1, and occludin (*p* < 0.01). *Butyricicoccus*, *Catabacter,* and *Eubacterium* were significantly positively correlated with occludin (*p* < 0.01). The expression levels of *Eubacterium* and *Catabacter* were positively correlated with those of ZO-1. *Clostridium_IV* was significantly negatively correlated with the intestinal barrier protein claudin-1 (*p* < 0.01) and positively correlated with the expression of ZO-1 and occludin (*p* < 0.05). *Clostridium_XlVa* had a strong positive correlation with the LPS, AST, and COX-2 levels (*p* < 0.01). *Aestuariispira* was positively correlated with DAO and D-lactic acid levels (*p* < 0.05).

## 4. Discussion

Chemotherapy remains one of the best treatments for cancer at present; however, the adverse reactions during treatment seriously affect the therapeutic effect and prognosis of patients with cancer and negatively affect their quality of life. Gastrointestinal toxicity is one of the most common and serious adverse reactions of chemotherapy, with the main manifestations being intestinal mucositis, diarrhea, and constipation [3,4,5]. Many studies have shown that intestinal flora may affect the efficacy and toxicity of chemotherapy drugs, and the imbalance in intestinal flora is related to the changes in intestinal mucosa and mucosal inflammation. Manipulation of the intestinal microbiota with antibiotics, probiotics, prebiotics, or fecal transplantation can help to improve the efficacy and reduce the toxicity of chemotherapeutic drugs [11,12]. Probiotics can increase the proportion of beneficial bacteria, regulate the balance of intestinal flora, reduce the levels of inflammatory factors, and restore the intestinal barrier [13,14]. Probiotics (as an adjuvant therapy) can improve adverse reactions in patients undergoing chemotherapy or radiotherapy [16,17,18]. With the development of high-throughput sequencing technology, the intestinal flora of healthy individuals at the phylum level is mainly composed of *Chlamydomonas*, *bacteria*, *Actinomycota, Proteus*, and *Clostridium* [19]. Changes in the abundance of *Proteus* spp. are considered a marker of an imbalance in the intestinal flora; however, the increased abundance of *Scleroderma* spp. and *Bacteroides* spp. can repair the structure of the intestinal flora [20,21,22].

Studies have shown that after treatment with chemotherapeutic drugs, the abundances of *Firmicutes* and *Bacteroides* are significantly reduced, whereas the abundances of *Proteus*, *Salviae*, *Gammaproteobacteria,* and *Verruca* are significantly increased [23,24]. In this study, we found that probiotic intervention improved the structure of the intestinal flora. Chemotherapeutic drugs can alter the structure of intestinal flora. Compared with the control group, the abundances of *Proteobacteria* and *Verrucomicrobia* in the experimental group were significantly increased. After intervention with probiotics, compared with the model group, the abundances of *Firmicutes*, *Bacteroidetes,* and *Actinobacteria* increased, but the abundances of *Proteobacteria* and *Verrucomicrobia* decreased. Chemotherapy treatment in mice results in dysbacteriosis, increased *E. coli*, *Clostridium*, and *Staphylococcus* abundance, and decreased *Bifidobacterium* and *Bacteroides* abundance [25]. After the rabbits were treated with 5-FU, the abundance of lactic acid bacteria and *Bacteroides* decreased, while that of *Escherichia, Clostridium,* and *Staphylococcus* increased [26]. *Shigella* is a Gram-negative bacterium that can aggravate intestinal damage and inflammatory factor expression [27]. Intervention with the composite probiotics increased the abundance of beneficial bacteria in the intestinal tract and reduced the abundance of potentially harmful bacteria [28]. At the genus level, we found that the increased abundance of *Escherichia/Shiga* in the experimental group indicated inflammation in the rabbits, and probiotic intervention significantly reduced this effect. The relative abundance of *Escherichia/Shigella* and *Sporobacter* in the experimental group was increased, while the relative abundance of *Bacteroides* was decreased. After the intervention with probiotics, the abundance of *Bacteroides*, *Blautia,* and *Bifidobacterium* in the probiotic intervention group was increased, while the abundance of *Escherichia/Shigella* and *Sporobacter* was decreased in the model group compared with that in the probiotic intervention group. In a previous meta-analysis, Touchefeu et al. found that patients treated with chemotherapy exhibited changes in the intestinal flora, most commonly a decrease in *Bifidobacterium*, *Clostridium XIVa,* and *Feces pulmonale* abundance, and an increase in *Enterobacteriaceae* and *Bacteroides* abundance [29]. *Clostridium rumen* promotes the development of the immune system by degrading polysaccharides to produce acetic acid and butyrate [30]. The relative abundances of *Enterobacteriaceae*, *Verrucomicrobiaceae*, *Bacteroidaceae*, *Clostridiaceae_1*, *Erysipelotrichaceae*, and *Desulfovibrionaceae* in the experimental group increased, while those of *Ruminococcaceae*, *Porphyromonadaceae*, and *Lachnospiraceae* decreased. The relative abundances of *Ruminococcaceae*, *Porphyromonadaceae*, *Lachnospiraceae*, *Actinobacteria*, *Bifidobacterium*, and *Prevotellaceae* were higher than those in the experimental group after the probiotic intervention. Moreover, the relative abundances of *Enterobacteriaceae*, *Verrucomicrobiaceae*, and *Desulfovibrionaceae* were reduced and were comparable to those in the control group. However, a significant increase in the abundance of *Bacteroidaceae* was observed in the probiotic intervention group.

The intestinal mucosal barrier comprises mechanical, chemical, immune, and biological barriers. The mechanical barrier is the most important barrier of the intestinal mucosa. The mucus layer and epithelial cells constitute the mechanical barrier, which prevents the invasion of harmful substances [31,32]. The integrity of the intestinal barrier is a marker of the balance of the intestinal ecosystem [33]. Tight-junction proteins, as important components of mechanical barriers, are composed of transmembrane proteins, including occlusive proteins (occludin), the claudin family, and the ZO-1 family. Occlusive proteins (occludin), zonuloccludin-1, and claudin-1 are the representative tight-junction proteins [34,35,36]. Studies have shown that regulating the intestinal flora and enhancing the gene expression of intestinal tight-junction proteins are potential mechanisms for repairing intestinal barriers [37]. Probiotics have been shown to improve the intestinal epithelial barrier function both in vitro and in vivo [12]. After chemotherapy, the intestinal permeability increases and an imbalance in the intestinal microflora affects intestinal permeability. Live probiotics can effectively reduce intestinal permeability and inflammation, and acute oral administration of *Bifidobacterium* to IL-10-deficient mice can acutely reduce colon permeability and increase the protein expression of ZO-1 and occludin [38]. Laval et al. found that *Lactobacillus rhamnosus* CNCM I-3690 partially restored the function of the intestinal barrier and increased the expression of occludin [39]. In another study, the oral administration of recombinant *Lactobacillus delbrueckii* CIDCA 133 reduced inflammatory infiltration and intestinal permeability, decreased the expression of the pro-inflammatory cytokine IL6, and increased the expression of IL-10, MUC2, and claudin-1 in the ileum [40]. The upregulation of the expression of claudin-1, ZO-1, and occludin has also been reported regarding oral probiotic recombinant strains [40]. *Bifidobacterium infantis* stabilizes claudins at tight junctions, improves intestinal permeability, and is associated with the internalization of claudin-4 and occludin [41]. The *Lactobacillus reuteri* Delbe strain CIDCA 133 has a beneficial effect on chemotherapy-induced intestinal damage by regulating the inflammatory pathway, improving epithelial barrier function, and improving intestinal permeability in mice. Upregulation of claudin-1 and downregulation of the NF-κB factor were observed [42]. *Bifidobacterium longum* DD98 can significantly reduce intestinal and liver toxicity and improve liver toxicity and biochemical indicators of oxidative stress. It can reduce the expression of pro-inflammatory cytokines and increase the expression of obtusin and ZO-1 [43]. In a previous study, the short-term oral administration of *Lactobacillus plantarum* WCSF1 to healthy volunteers increased the expression of the occlusive protein and ZO-1 genes [44]. *Bifidobacterium infantis* and *Lactobacillus acidophilus* can normalize occludin and claudin-1 protein expression [44]. This is consistent with our study. We detected the protein expressions of claudin-1, ZO-1, and occludin in the rabbit ileum and the protein expressions of claudin-1, ZO-1, occludin, and MUC2 in the serum using ELISA. In the intestinal tissue, compared with the control group, the protein expression levels of claudin-1, ZO-1, and occludin in the model group were significantly reduced. After the probiotic intervention, the expression levels of claudin-1, ZO-1, occludin, and MUC2 were significantly increased, without significant differences from the control group and with significant differences from the model group. In the serum, compared with the control group, the expression levels of claudin-1, ZO-1, and occludin in the model group were significantly reduced, and were restored after being regulated by probiotics. The expression of the MUC2 gene decreased in the model group, and the serum expression of the MUC2 gene protein increased after probiotic intervention. We found that changes in the permeability of the intestinal barrier could be determined by measuring the levels of DAO and D-lactic acid in the serum. Compared with the control group, the DAO and D-lactic acid levels in the experimental group were significantly increased. Compared with the control group, the levels of DAO and D-lactic acid in the model group were also significantly higher. After probiotic regulation, the levels of DAO and D-lactic acid decreased, which may have repaired the intestinal barrier to some extent.

The NF-κB pathway is a classical inflammatory signaling pathway that plays an important role in the process of intestinal injury [45]. Abnormal activation of immune cells in the intestinal tract leads to the excessive production of pro-inflammatory cytokines, resulting in intestinal inflammation [46]. TNF-α is a pro-inflammatory factor that can interact with other cytokines to induce the release of inflammatory mediators in the body [47]. Probiotic *Bifidobacterium infantis* can effectively reduce the expression of NF-κB and pro-inflammatory factors in rat intestinal mucositis caused by chemotherapy [48]. A study showed that probiotic DM#1 ameliorated chemotherapy-induced intestinal mucosal injury in rats, reduced pro-inflammatory cytokine levels and neutrophil infiltration, and improved intestinal permeability [17]. In another study, treatment with *Bifidobacterium infantis* and *Bacteroides* spp. reduced NF-κB activation, endotoxin levels, and pro-inflammatory cytokines [26]. *Bifidobacterium bifidum* G9-1 intervention can also reduce the concentrations of pro-inflammatory cytokines TNF-α and IL-1β [49]. *Lactobacillus plantarum* LC27 and *Bifidobacterium longum* LC67 can inhibit the growth of *Escherichia coli* and the LPS-mediated activation of NF-κB [50]. *Bifidobacterium infantis* and *Lactobacillus acidophilus* can reduce the activation of NF-κB in Caco-1 cells induced by the inflammatory factor IL-1β to protect the intestinal barrier from IL-2β stimulation [44]. We detected the inflammatory factors LPS, TNF-α, and IL-6 in the serum of rabbits after local hepatic perfusion chemotherapy with the intervention of viable *Bifidobacterium* tetralogy. The results of this study confirmed that compared with the control group, the serum levels of LPS, TNF-α, and IL-6 of rabbits in the experimental group were significantly increased. Compared with the experimental group, the expression levels of serum TNF-α and IL-6 after regulation in the probiotic group were significantly reduced, making them comparable to the levels in the control group. After regulation, the expression levels of LPS and NF-κB in serum of the probiotic group were decreased, and they were all higher than those in the control group. This indicated that the viable probiotic *Bifidobacterium* tetralogy could inhibit the NF-κB signaling pathway, reducing the expression of intestinal inflammatory factors in rabbits. *Bifidobacterium longum* DD98 can significantly reduce intestinal and liver toxicity and alleviate liver toxicity and oxidative stress [43]. The oral administration of *Lactobacillus plantarum* LC27 and *Bifidobacterium longum* LC67 in mice with hepatic steatosis can reduce the ALT and AST levels in the blood and liver [51]. The probiotic complex maintains the integrity of the intestinal barrier, blocks lipopolysaccharide (LPS) translocation, and inhibits the activation of the TLR4/NF-κB pathway and the production of inflammatory factors in the liver and ileum [28]. Compared with the control group, the serum ALT, AST, NLRP3, and p-STAT3 in the experimental group increased significantly, on average. The expression levels of ALT and p-STAT3 in the rabbit serum of the probiotic group were significantly decreased after intervention, whereas the expression levels of ALT and NLRP3 in the rabbit serum of the probiotic group were decreased after intervention, which were all higher than those in the control group.

We found a significant positive correlation between *Firmicutes* and ZO-1 and occludin expression at the phylum level. We also observed a significant positive correlation between *Candidatus_Saccharibacteria* and ZO-1 and occludin expression and a positive correlation between *Candidatus_Saccharibacteria* and claudin-1 expression. *Proteobacteria* had a significant positive correlation with the levels of the inflammatory factors (LSP and TNF-α) and liver injury factors (AST, ALT, and STAT3), and a positive correlation with the levels of DAO and IL-6. At the genus level, the harmful bacterium *Escherichia/Shigella* had a significant positive correlation with the levels of LSP and TNF-α and a positive correlation with the levels of AST, ALT, and COX-2. However, it was significantly negatively correlated with the expression of claudin-1, ZO-1, and occludin. *Lachnospiraceae_incertae_sedis* was significantly positively correlated with the expression of ZO-1 and occludin. This indicates that, in general, the abundance of beneficial bacteria was significantly positively correlated with the expression of the intestinal tight-junction proteins, and that of harmful bacteria was significantly correlated with the expression of the inflammatory and liver injury factors.

## 5. Conclusions

Our research showed that the intestinal flora of rabbits with local liver perfusion chemotherapy was disordered, the relative abundance of beneficial bacteria, such as *Firmicutes* and *Bacteroides*, decreased, and the relative abundance of harmful bacteria, such as *Proteus*, increased. The administration of quadruple viable bacteria of *Bifidobacterium* regulated the intestinal flora disorder caused by local hepatic perfusion chemotherapy, and the relative abundance of beneficial bacteria, such as *Bifidobacterium, Ruminococcus*, *Prevost’s* bacteria, and *Blautia*, inhibited the relative abundance of harmful bacteria, such as *Proteus*, *Verruca*, *Escherichia/Shigella,* and *Sporobacter.* There was no significant difference in the α diversity in the microbial communities in each group. There were, however, significant differences in β diversity in the microbial communities in each group. This study also showed that live *Bifidobacterium* tetralogy can reduce the levels of proinflammatory cytokines and liver injury factors in serum of rabbits undergoing local hepatic perfusion chemotherapy. It also significantly increased the protein expression levels of claudin-1, ZO-1, and occludin genes in rabbit serum and ileum tissue and improved the permeability of the intestinal barrier.

## Figures and Tables

**Figure 1 biology-13-00327-f001:**
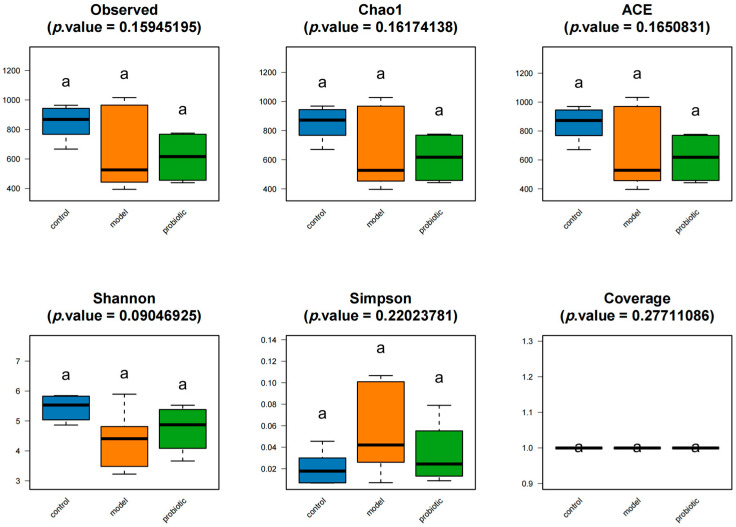
We analyzed the alpha diversity of the control group, model group, and probiotic intervention group using the Observed, Chao, ACE, Shannon, Simpson, and Coverage indices. In the figure, “a” indicates that there was no statistical difference between the groups. The Observed index was used to estimate the number of OTU in each group. The ACE index was used to estimate the number of OTU in each group of intestinal flora. Coverage refers to the coverage of each group (clone) library. The higher its value, the higher the probability that the sequences in each group will be detected, reflecting the real situation of microorganisms in each group. The Chao1 algorithm estimates the index of the number of OTU in each group. The more OTU, the more species in this group. The Shannon index was used to estimate one of the microbial diversity indexes in each group. The Simpson diversity index is also often used to reflect the alpha diversity index. The greater the Shannon value, the higher the diversity of microbial communities in each group, which is contrary to the Simpson value.

**Figure 2 biology-13-00327-f002:**
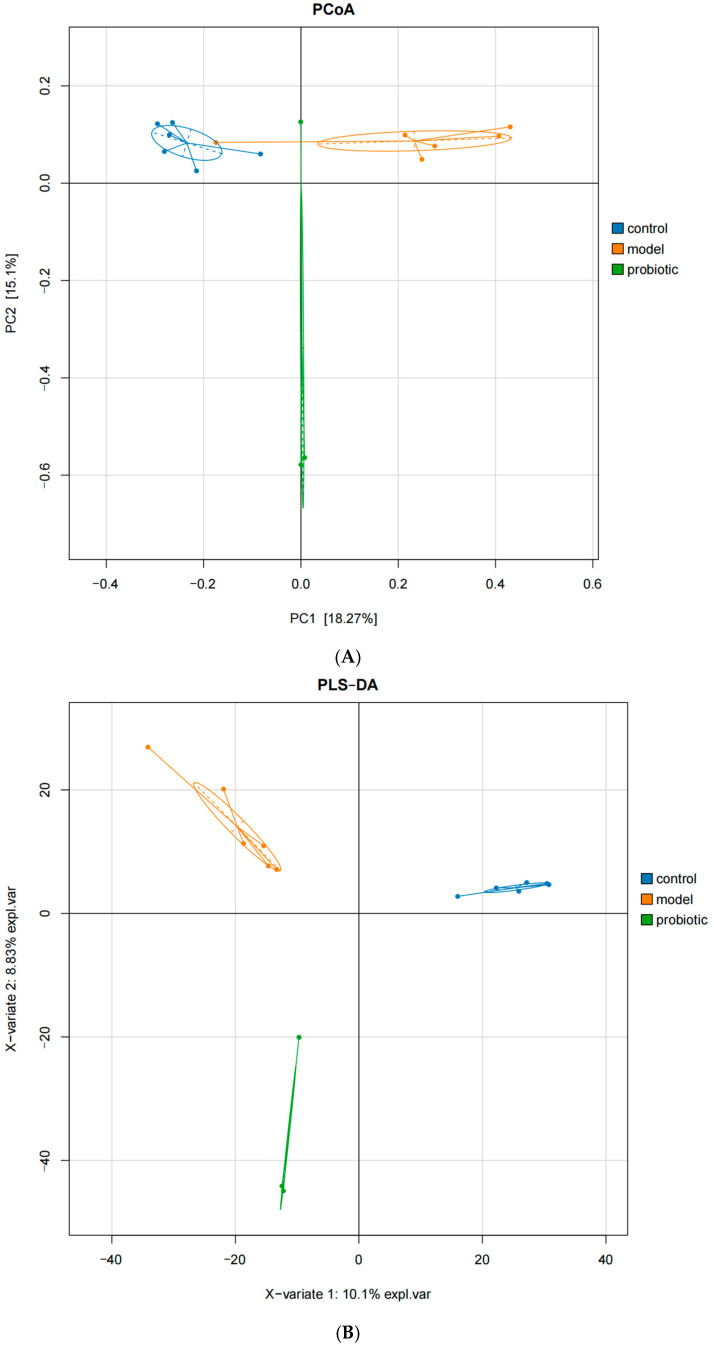
Differences in β diversity between groups. (**A**) PCoA is a visualization technology, which expresses the similarity of research data through coordinate visualization. This is an unconstrained dimension reduction analysis method, which is usually used to study the similarity or difference in sample community composition in each group. (**B**) Partial least squares discriminant analysis (PLS-DA) can effectively distinguish each group of measurement data points by properly rotating the principal component and can be used to identify the influencing variables that may lead to differences between groups. Each group of samples is represented by different colors.

**Figure 3 biology-13-00327-f003:**
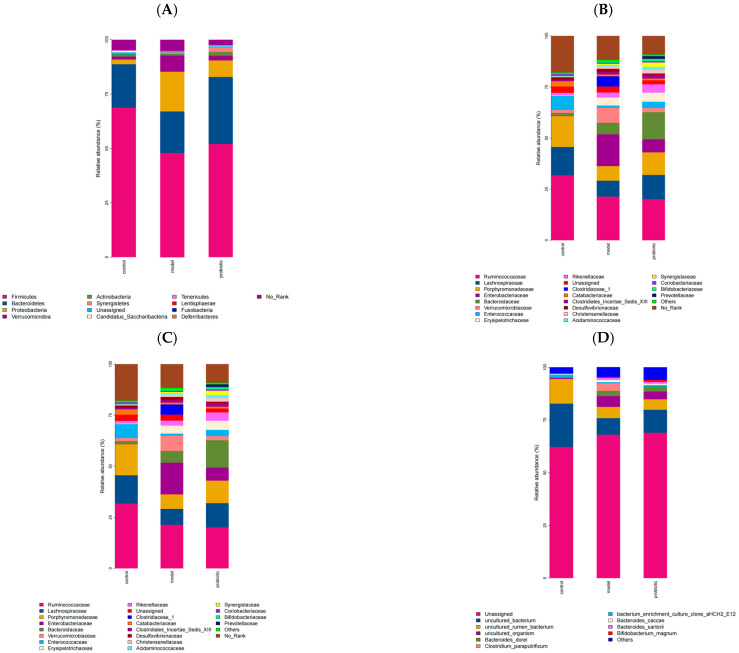
Differences in the structural compositions of the intestinal flora among different groups. (**A**) The relative abundance changes in the intestinal flora in each group at the phylum level. (**B**) The relative abundance changes in the intestinal flora in each group at the family level. (**C**) The relative abundance changes in the intestinal flora in each group at the genus level. (**D**) The relative abundance changes in the intestinal flora in each group at the species level.

**Figure 4 biology-13-00327-f004:**
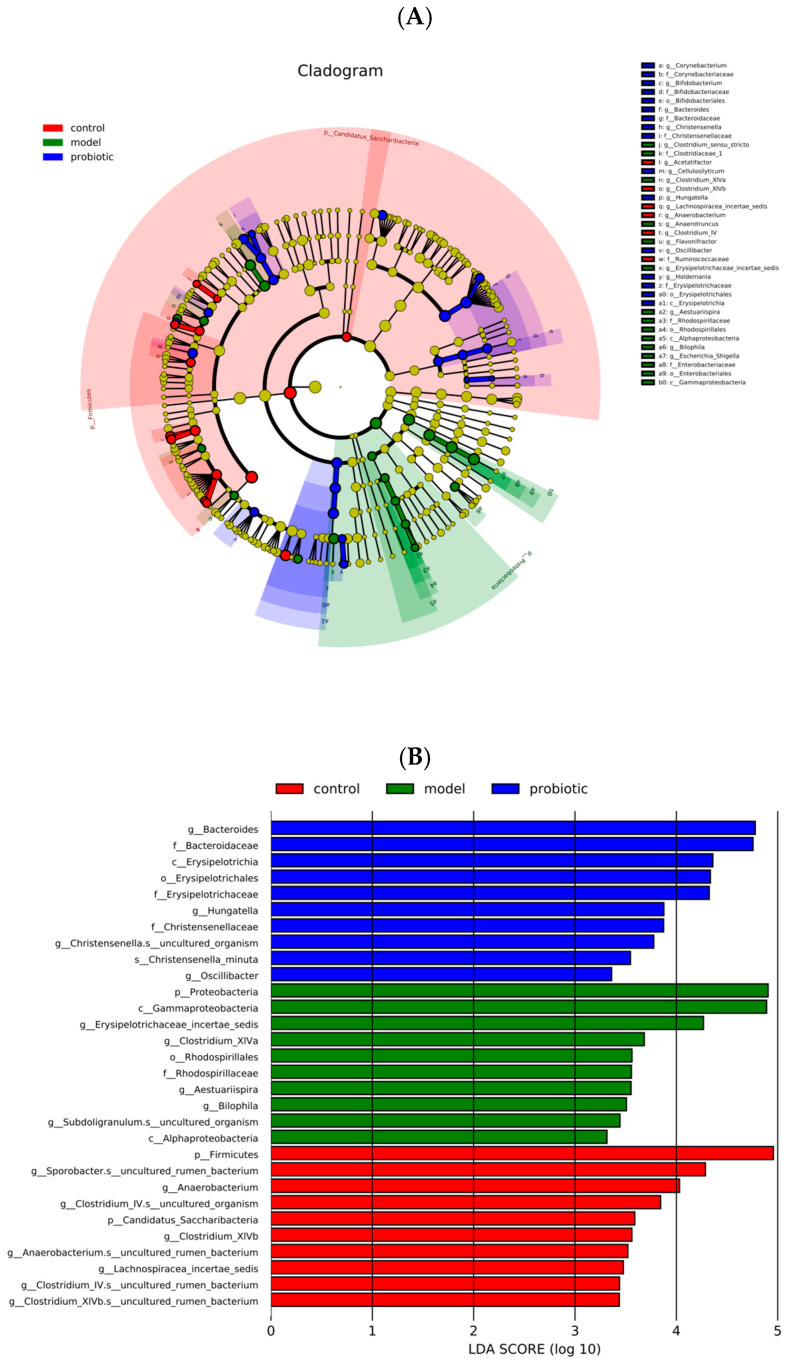
Potential biomarkers representing the differences between groups. (**A**) LEfSe analysis provides the phylogenetic distribution differences from phylum to species in each group. (**B**) The LDA score was used to identify the different bacteria between groups. By selecting the top 50 species with the smallest *p*-value, we used |LDA| > 2 and the *p*-value to compile an evolutionary bifurcation diagram to screen the species with significant differences. First, we selected the 10 species with the smallest *p*-value in each group to draw the bar chart shown here. Different species with LDA value (log 10) > 2 in different taxa, and species with significantly higher abundance in this taxa, are represented by bar charts with different colors. The size of the LDA score is represented by the length of the bar chart.

**Figure 5 biology-13-00327-f005:**
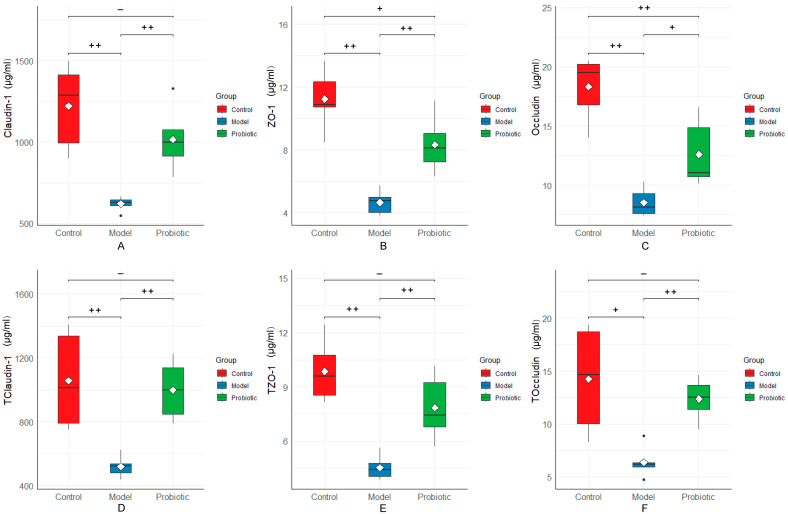
Expression of intestinal barrier tight-junction proteins in the serum and colon tissues among the different groups. We performed statistical analysis using the box diagram. (**A**) Expression of claudin-1 in the serum. (**B**) Expression of ZO-1 in the serum. (**C**) Expression of occludin in the serum. (**D**) Expression of claudin-1 in the colon tissues. (**E**) Expression of TZO-1 in the colon tissues. (**F**) Protein expression of occludin in the colon tissues. In this analysis, the Kaplan–Meier method was used to plot the survival curve, and the significance was determined using the log-rank test. The difference was assessed using the Tukey test after one-way ANOVA: “++” indicates *p* < 0.01 and “+” indicates *p* < 0.05. Here, “+” means that the difference was statistically significant, and “−“ means that the difference was not statistically significant.

**Figure 6 biology-13-00327-f006:**
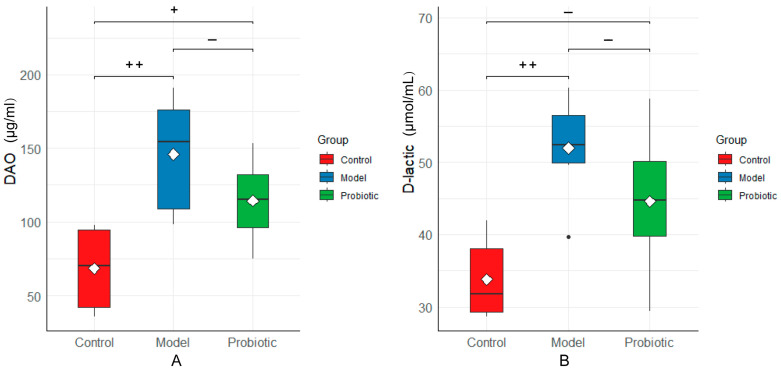
DAO and D-lactic acid levels (indicators of intestinal barrier permeability) between the groups. We performed statistical analysis using the box diagram. (**A**) DAO levels in the serum. (**B**) D-lactic acid levels in the serum. In this analysis, the Kaplan–Meier method was used to plot the survival curve, and the significance was determined using the log-rank test. The difference was assessed using the Tukey test after one-way ANOVA: “++” indicates *p* < 0.01 and “+” indicates *p* < 0.05. Here, “+” means that the difference was statistically significant, and “−“ means that the difference was not statistically significant.

**Figure 7 biology-13-00327-f007:**
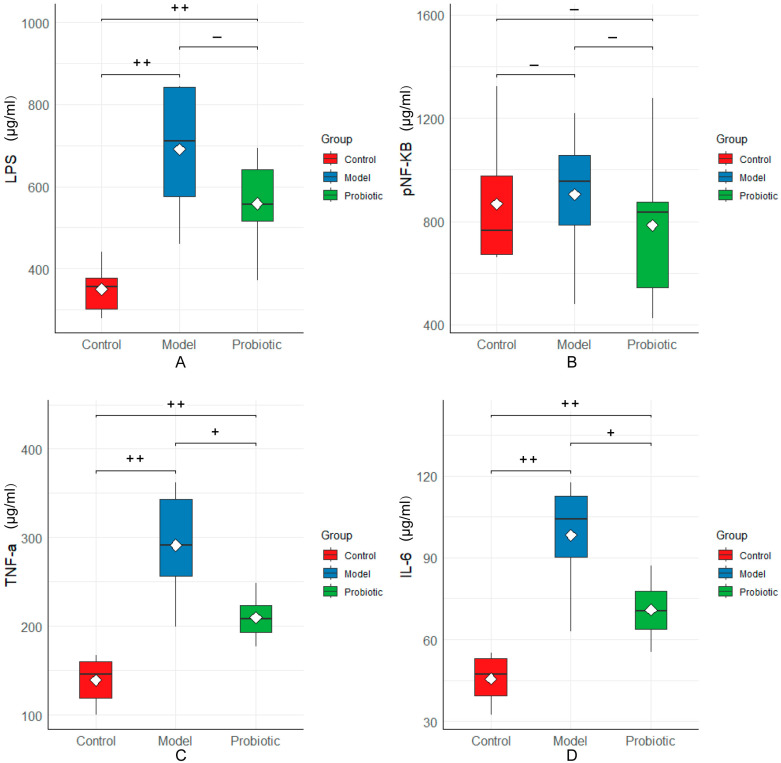
Levels of the serum inflammatory factors among the groups. We performed statistical analysis using the box diagram. (**A**) LPS levels in the serum. (**B**) NF-κB levels in the serum. (**C**) Serum TNF-α levels. (**D**) IL-6 levels in the serum. In this analysis, the Kaplan–Meier method was used to plot the survival curve, and the significance was determined using the log-rank test. The difference was assessed using the Tukey test after one-way ANOVA: “++” indicates *p* < 0.01 and “+” indicates *p* < 0.05. Here, “+” means that the difference was statistically significant, and “−“ means that the difference was not statistically significant.

**Figure 8 biology-13-00327-f008:**
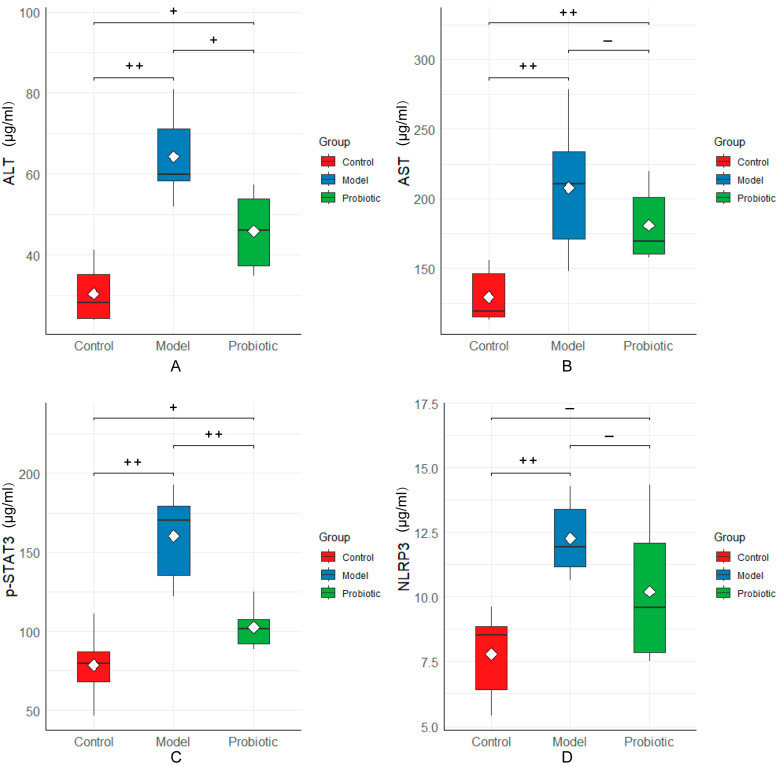
Levels of the inflammatory factors of liver injury among the groups. We performed statistical analysis using the box diagram. (**A**) ALT levels in the serum. (**B**) AST levels in the serum. (**C**) Serum STAT3 levels. (**D**) NLRP3 levels in the serum. In this analysis, the Kaplan–Meier method was used to plot the survival curve, and the significance was determined using the log-rank test. The difference was assessed using the Tukey test after one-way ANOVA: “++” indicates *p* < 0.01 and “+” indicates *p* < 0.05. Here, “+” means that the difference was statistically significant, and “−“ means that the difference was not statistically significant.

**Figure 9 biology-13-00327-f009:**
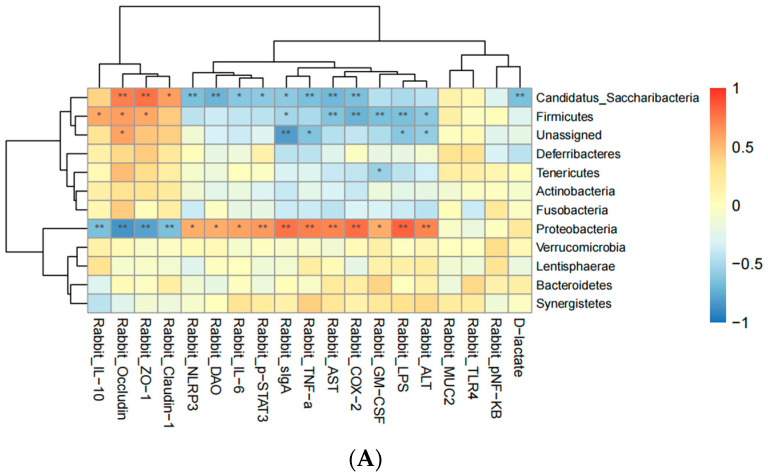
Heat map of the correlation between the intestinal flora and clinical inflammatory factors in each group. We analyzed the correlation between the intestinal flora and clinical inflammatory factors by screening the differential intestinal flora. Red represents a positive correlation, while blue represents negative correlation. (**A**) Correlation between the phylum level of the intestinal microflora and clinical inflammatory factors in each group. (**B**) Correlation between the genus level of the intestinal flora and clinical inflammatory factors in each group. In the figure, “**” indicates *p* < 0.01, and “*” indicates *p* < 0.05.

## Data Availability

The data presented in this study are available upon request from the corresponding author.

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
