# Peer review of "Effects of Combined Live Bifidobacterium, Lactobacillus, Enterococcus, and Bacillus Cereus Tablets on the Structure and Function of the Intestinal Flora in Rabbits Undergoing Hepatic Artery Infusion Chemotherapy"

_biology, 2024, doi:10.3390/biology13050327_

Round 1

Reviewer 1 Report

Comments and Suggestions for Authors

The study in question delves into the biological mechanisms by which probiotics can mitigate the adverse effects of chemotherapy, specifically through the regulation of intestinal flora. The research focuses on the impact of a combination of probiotics—Bifidobacterium, Lactobacillus, Enterococcus, and Bacillus Cereus—on the intestinal microbial structure and barrier function in rabbits subjected to local hepatic chemotherapy infusion. The findings suggest that this probiotic blend not only helps in balancing the intestinal flora by boosting beneficial bacteria and suppressing harmful ones but also plays a role in reducing inflammation and liver damage caused by chemotherapy. This could potentially pave the way for using probiotics as an adjunct therapy to lessen the negative reactions associated with chemotherapy, offering a promising avenue for improving patient outcomes during cancer treatment. The implications of such research are significant, as they offer a non-invasive method to enhance the quality of life for patients undergoing rigorous cancer therapies. However, some questions need to be addressed in this study:

Methods:

Why did you choose 0.75 mg/bid dose? And why for 7 days? 

Are these probiotics commercially available?

Can chemotherapy destroy live bacteria in probiotics? 

Comments on the Quality of English Language

Simple Summary: Lines 16-20: long sentence and hard to understand.

“In this study, by establishing a rabbit model of local hepatic perfusion chemotherapy, we studied the influence of live bifidobacterium tetralogy on the intestinal flora structure and intestinal barrier function of local hepatic arterial perfusion chemo therapy rabbits, and explored the possible potential mechanism, so as to regulate the intestinal flora for probiotics and reduce the side effects of drugs after local hepatic perfusion chemotherapy. “

Author Response

Thank you very much for taking the time to review this manuscript. Please find the detailed responses below and the corresponding revisions in track changes in the re-submitted files.I will highlight the modified part in red font.

The study in question delves into the biological mechanisms by which probiotics can mitigate the adverse effects of chemotherapy, specifically through the regulation of intestinal flora. The research focuses on the impact of a combination of probiotics—Bifidobacterium, Lactobacillus, Enterococcus, and Bacillus Cereus—on the intestinal microbial structure and barrier function in rabbits subjected to local hepatic chemotherapy infusion. The findings suggest that this probiotic blend not only helps in balancing the intestinal flora by boosting beneficial bacteria and suppressing harmful ones but also plays a role in reducing inflammation and liver damage caused by chemotherapy. This could potentially pave the way for using probiotics as an adjunct therapy to lessen the negative reactions associated with chemotherapy, offering a promising avenue for improving patient outcomes during cancer treatment. The implications of such research are significant, as they offer a non-invasive method to enhance the quality of life for patients undergoing rigorous cancer therapies. However, some questions need to be addressed in this study:

Methods:

 1.Why did you choose 0.75 mg/bid dose? And why for 7 days? 

Response 1:We used Combined Live Bifidobacterium, Lactobacillus, Enterococcus and Bacillus Cereus Tablets of 0.75 mg/bid dose by reviewing the references and strictly following the instructions of the probiotic formulation.Because clinically, patients with liver cancer are generally discharged from the hospital within 7 days after hepatic artery embolization and chemotherapy perfusion. To simulate the effect of acute probiotic intervention on gut microbiota and its intestinal barrier function after hepatic arterial perfusion chemotherapy, we used a probiotic preparation for a 7-day intervention.

  1. Are these probiotics commercially available?

Response 2:Thank you very much for your comments. This probiotic preparation is sold in the market by specialized probiotic preparation companies.

  1. Can chemotherapy destroy live bacteria in probiotics? 

Response 3:Thank you very much for your valuable comments. We found that the relative abundance of Bifidobacterium species increased after intervention with probiotic preparations. But we didn't find that chemotherapy destroyed the live bacteria in the probiotics. We found that chemotherapy decreased the relative abundance of probiotics and increased the relative abundance of harmful bacteria through literature review and in conjunction with this study. Our research focuses on whether the intervention of probiotic preparations can improve the intestinal microbiota structure and intestinal barrier of hepatic arterial perfusion chemotherapy drugs.

Simple Summary: Lines 16-20: long sentence and hard to understand.

“In this study, by establishing a rabbit model of local hepatic perfusion chemotherapy, we studied the influence of live bifidobacterium tetralogy on the intestinal flora structure and intestinal barrier function of local hepatic arterial perfusion chemo therapy rabbits, and explored the possible potential mechanism, so as to regulate the intestinal flora for probiotics and reduce the side effects of drugs after local hepatic perfusion chemotherapy.”

Response 4:Thank you very much for your valuable comments, and I have briefly summarized this sentence and marked in red font in the article. In this study, a rabbit model of liver local arterial perfusion chemotherapy was established. The effects of Combined Live Bifidobacterium, Lactobacillus, Enterococcus and Bacillus Cereus Tablets on the structure of intestinal microbiota and intestinal barrier function in rabbits treated with local arterial perfusion chemotherapy were studied, and the possible potential mechanisms were discussed. This study provides a theoretical basis for probiotics to regulate intestinal microbiota and improve the side effects of local arterial perfusion chemotherapy drugs in the liver.

Reviewer 2 Report

Comments and Suggestions for Authors

I would thank the editor for the possibility to revise this manuscript, the scientific background under the proposed hypothesis is well documented and described by the authors and the experimental procedures applied to verify the hypothesis has been appropriate. Furthermore, the bioinformatics analysis is very well performed.

In this manuscript the authors contribute to shed light on use of probiotics as a potential alternative therapeutic strategy to prevent the adverse reactions caused by chemotherapy with local hepatic perfusion.  In particular, the authors investigate the effects of Combined Live Bifidobacterium, Lactobacillus, Enterococcus and Bacillus Cereus Tablets on the intestinal microbial structure and intestinal barrier function and its potential mechanism in rabbits after local hepatic chemotherapy infusion.

I have only few minor clarifications/changes to request that could improve the manuscript.

Q1: In the Simple Summary define the acronym HCC (page 1, line 14).

Q2:  The scientific background on the function of the intestinal flora composition in the chemotherapy/immunotherapy efficacy and toxicity drugs is well described. Although, in the introduction, the description of how specific probiotics intervene in the drug’s adverse events could be expanded. I think this in-depth analysis can help readers to understand the chosen probiotic formulation.

Q3:  In the Experimental Operation of the Materials and Methods section the authors indicate that the experimental group (n=12 rabbits) was randomly divided into three groups. Actually, the experimental group seems to be divided into two groups: one treated only with chemotherapy and the other with the chemotherapy + probiotics. Furthermore, the two terms “model” and “probiotic” used in the figures should be reported in this section. I think that the authors should be clarify this point.

Q4: As reported in the literature, it is known that the choice of the 16S regions affects the identification of the bacterial community potentially identify (PMID: 30720800). Can the authors clarify the choice to analyze only the V3-V4 hypervariable region of the 16S gene?

Author Response

Thank you very much for taking the time to review this manuscript. Please find the detailed responses below and the corresponding revisions in track changes in the re-submitted files.I will highlight the modified part in red font.

Q1: In the Simple Summary define the acronym HCC (page 1, line 14).

Response 1: Thank you very much for your valuable comments. I'll mark the changes in red. It was my negligence not to use the full spelling of HCC.

Q2:  The scientific background on the function of the intestinal flora composition in the chemotherapy/immunotherapy efficacy and toxicity drugs is well described. Although, in the introduction, the description of how specific probiotics intervene in the drug’s adverse events could be expanded. I think this in-depth analysis can help readers to understand the chosen probiotic formulation.

Response 2: Thank you very much for this valuable comment. Our current focus is on the study of the effect of probiotic intervention on hepatic arterial perfusion chemotherapy, gut microbiota and intestinal barrier. The impact of the specific probiotic flora you proposed is the next step for our team's research. Thank you again for your valuable comments.

Q3:  In the Experimental Operation of the Materials and Methods section the authors indicate that the experimental group (n=12 rabbits) was randomly divided into three groups. Actually, the experimental group seems to be divided into two groups: one treated only with chemotherapy and the other with the chemotherapy + probiotics. Furthermore, the two terms “model” and “probiotic” used in the figures should be reported in this section. I think that the authors should be clarify this point.

Response 3: Thank you very much for your opinion. This is my negligence. After the successful modeling, the model group was randomly divided into two groups. One group is the model group, that is, the hepatic artery local perfusion chemotherapy group, and the other group is the probiotic intervention group, that is, the hepatic artery local perfusion chemotherapy+Combined Live Bifidobacterium, Lactobacillus, Enterococcus and Bacillus Cereus Tablets group. In the article, I made changes through red comments.One group was model group: liver local chemotherapy perfusion group (n=6). In this paper, it is collectively called model group, and in the figure, model group is uniformly used for labeling. Another group of probiotics intervention group: liver local chemotherapy and perfusion combined with live Bifidobacterium, Lactobacillus, Enterococcus and Bacillus cereus tablets (n=6). In this paper, it is collectively referred to as the probiotic intervention group, and in the figure, the probiotic group is uniformly labeled.Thank you again for your valuable advice.

Q4: As reported in the literature, it is known that the choice of the 16S regions affects the identification of the bacterial community potentially identify (PMID: 30720800). Can the authors clarify the choice to analyze only the V3-V4 hypervariable region of the 16S gene?

Response 4: Thank you very much for your valuable advice. By consulting the literature and combining the microbial analysis methods of several teams. It is found that the most common experimental sequencing method for sequencing intestinal flora at present mainly highlights the selection of V3-V4 hypervariable region of 16S rRNA, which can reflect the changes of intestinal flora structure. Therefore, we used 16S rRNA sequencing to study the intestinal flora of rabbits.